# Decision-Making in Patients with Vasovagal Syncope: A Preliminary Study

**DOI:** 10.3390/biology12070930

**Published:** 2023-06-29

**Authors:** Muriel Méchenin, Jacques-Olivier Fortrat

**Affiliations:** Equipe CarMe, MITOVASC, SFR ICAT, CNRS, INSERM, Médecine Vasculaire, CHU Angers, Faculty of Medicine, Université d’Angers, 49933 Angers, France; Muriel.Mechenin@chu-angers.fr

**Keywords:** autonomic nervous system, baroreflex, decision-making, orthostatic intolerance, syncope, vasovagal syncope

## Abstract

**Simple Summary:**

This study looked at how people with syncope, mainly due to blood phobia or mainly occurring during prolonged standing positions, perform a task that measures their decision-making ability. Out of 332 young participants, 98 were in a control group, 10 in the blood phobia group, and 38 in the position group. The participants performed a computer task to see how well they could handle uncertainty when choosing options. The results showed that people with blood phobia took longer to make decisions in uncertain situations compared to the control and position groups. These results suggest that blood phobia and position syncope are separate conditions and that decision-making can be targeted in the management of patients with blood phobia.

**Abstract:**

The aim of this study was to evaluate the differences in performance during a decisional conflict task between subjects with emotional/blood phobia and those with an orthostatic vasovagal syncope. A total of 332 young subjects were included, from which 99 were excluded because of their condition or treatment. The subjects were classified into four groups depending on their responses to a questionnaire: 98 in a control group, 10 in an emotional/blood phobia syncope group, 38 in an orthostatic syncope group, and 87 in an unclear status group. This former group was excluded. The subjects performed a decisional conflict task to quantify their conflict-management ability. The task was the computer version of the Simon Task. Emotional/blood phobia syncope subjects showed a delayed reaction time when faced with decisional conflict in comparison with the control and orthostatic syncope subjects (55.8 ± 17.7 ms, 20.5 ± 4.9 ms, and 13.4 ± 9.2 ms, respectively, *p* ≤ 0.05). Our result suggests that emotional/blood phobia and orthostatic syncope are two clinical entities. Decisions could be a target of management in patients with emotional/blood phobia syncope. The altered decision-making of subjects with emotion/blood phobia syncope emphasized the role of higher cerebral functions in blood pressure control.

## 1. Introduction

Vasovagal syncope has recently been recognized as a major medical problem because of its relatively high frequency, challenging management, social costs, and its impact on quality of life [1,2,3,4,5]. Many recent efforts have thus been made to better understand it. Interestingly, vasovagal syncope remains widely considered as a single entity, while the trigger can be mainly emotional or mainly orthostatic [2,6,7]. Orthostatic vasovagal syncope occurs during a prolonged standing position [4,8]. A standing position puts the voluminous human brain above the heart while venous blood volume is pooled below the diaphragm. The cardiac pre-load decreases, and blood pressure homeostasis is challenged. The baroreflex operates on a beat-to-beat basis to maintain blood pressure during such challenges, and, in this context, it increases the heart rate and induces vasoconstriction [9]. In addition to the standing position, another important factor precipitating vasovagal syncope is heat exposure [3]. In such situations, skin vasodilation dissipates body heat but, in doing so, also pools blood away from the thorax. A paradoxical situation arises when standing in a warm environment, which requires both vasoconstrictions to maintain blood pressure and vasodilation to improve heat exchange. In such a context, efficient cardiovascular regulation requires correct and fast decision-like processes to be made between vasodilation and vasoconstriction. There is still uncertainty as to whether changes in decision-making could increase the likelihood of syncope. While there are multiple tools to assess decision-making, they often concentrate on the cognitive ability to recognize and select options. The Simon task, which measures reaction times, is more relevant for evaluating physiological regulation [10]. We hypothesized that the Simon task could distinguish between emotional/blood phobia and orthostatic syncope, with patients in the orthostatic subtype displaying less efficient decision-making in comparison with those for whom it is emotional/blood phobia.

## 2. Materials and Methods

### 2.1. Questionnaire

The target population was young and healthy individuals in order to minimize the risk of including those with a cardiac, neurologic, or situational syncope. Individuals aged 18 years or older were approached at various locations within the University of Angers and were asked if they would be willing to participate in the research, which involved answering a short questionnaire and playing a brief computer game, taking approximately 10 min in total. The questionnaire was designed to achieve the study’s objectives and not to propose a new medical tool. The experiment was conducted over five-week campaigns spanning five years. The first two campaigns were utilized to refine the questionnaire and optimize the inclusion strategy. Through these initial trials, we discovered that inclusions decreased significantly after one week at a single location and after five weeks when considering the entire university. Additionally, we found that we could expect approximately 120 subjects per campaign. Some subjects wanted to perform the test multiple times, and, in such cases, age, weight, and height were set to 0 for easy data discard. The three inclusion campaigns were conducted between 2015 and 2017. The questionnaire, which can be found in Appendix A, served to identify individuals older than 30 years old and those with health issues and to exclude them from the analysis. Subjects were considered to have a health issue if they reported needing medication or if the issue required regular visits to a doctor. Subjects who had taken medication on the day of the experiment were also excluded. The specific types of medication or health issues were not specified. Subjects with any health issues were excluded to minimize the risk of including syncope cases that were unrelated to vasovagal episodes. Additionally, subjects who were taking medication were excluded to minimize the risk of including any side effects that could manifest as dizziness or loss of consciousness. Subjects with a medical history that included a transient loss of consciousness but did not require follow-up or regular visits to a doctor were included. Another section of the questionnaire aimed to determine a summary of the subject’s history of a transient loss of consciousness. Based on their responses, participants were classified into three main groups: “syncope subjects”, who reported experiencing at least three spells; “control subjects”, who reported no spells; and “unclear,” the remaining participants. The “unclear” group was then excluded from the analysis. The “syncope subjects” were further divided into two subgroups based on the primary trigger: emotional/blood phobia or orthostatic syncope. Participants did not complete the questionnaire themselves but answered one of the authors, who provided clarification if needed. Measurements such as blood pressure could not be performed due to the inability to reach a consensus on the measurement conditions during this in-field questionnaire. Additionally, administrative and human experiment issues also contributed to this limitation. Although a tendency toward low blood pressure was reported, the clinical examination of patients with vasovagal syncope is typically uninformative [2,7,11]. Furthermore, blood pressure is expected to be within the normal range for this population of young people with no health issues who receive medical care from the university. All subjects gave their written informed consent, and the Angers ethics committee approved the protocol, which was performed in accordance with the Declaration of Helsinki.

### 2.2. Decisional Conflict Test

A computer version of the Simon task was proposed to the subjects following the questionnaire. This test explores the response selection stage of decision-making and has been extensively described elsewhere [10,12]. Shortly after an 800 ms fixation cross in the middle of the computer screen followed by a 250 ms empty screen, a red or a blue square appeared on the left or the right side of the screen for a maximum of 1000 ms. Subjects were instructed to press a left-side key for a blue square or a right-side key for a red square, whether the square appeared on the left or on the right side of the screen (Q key and M key on a French keyboard, respectively). The Simon task involved a confusing stimulus when the square was presented on the opposite side of the associated response key. It also involved a non-confusing stimulus when the square and the associated response key were on the same side. Confusing and non-confusing stimuli were termed incongruent and congruent stimuli, respectively. Twenty-eight squares were successively presented at random during the test. Half of the trials were incongruent. A practice trial was performed, as previously described [10]. The reaction time was measured from the onset of the stimulus to the subject response. The mean reaction time for congruent and incongruent stimuli was obtained, and the Simon effect, defined as the difference between these two mean times, was then determined. Accuracy was determined as the percentage of correct responses.

### 2.3. Statistics

Determining the sample size for this study was challenging due to the lack of comparable data in the literature. However, Bialystok et al. obtained significant results in groups of 10 subjects using the Simon task [10]. Additionally, the sample size could be estimated using epidemiological data from a comparable population [13]. Based on these data, we expected to enroll approximately 39% of subjects with a syncope, with 64% of them experiencing recurrence and 39% with emotional/blood phobia. Using these estimates, we anticipated that approximately 9.8% of subjects had a recurrent emotion/blood phobia syncope, and 10.2% had a recurrent orthostatic syncope. In theory, the inclusion of just over 100 subjects would have been sufficient to form groups of at least 10 subjects. However, due to the design of the experiment and anticipated exclusions, we increased the number to 250 subjects. The data are presented as means ± SEM. The control, emotional/blood phobia syncope, and orthostatic syncope subjects were compared by ANOVA and post hoc t-tests with Bonferroni correction. The sex ratios of these three groups of subjects were compared by Fisher’s exact tests. Data analysis was performed using Prism software (Prism 5.01, GraphPad Software, San Diego, CA, USA). Statistical significance was set at *p* ≤ 0.05.

## 3. Results

Of the 332 subjects who participated in the study, 99 were excluded due to being over 30 years of age, having a health issue, or taking medication on the day of the questionnaire. Among the remaining subjects, 78 were classified as “unclear” because they reported experiencing only one or two episodes of a transient loss of consciousness. In total, 98 were classified as control subjects (42 of them were female; 22 ± 2 y, 22 ± 2 kg/m²; mean age, mean body mass index (BMI)), and 57 as syncope subjects (36 of them were female; 22 ± 2 y, 21 ± 3 kg/m²). Within this latter group, we identified 10 emotional/blood phobia syncope subjects (5 of them were female; 21 ± 2 y, 21 ± 2 kg/m²) and 38 orthostatic syncope subjects (25 of them were female; 22 ± 2 y, 21 ± 3 kg/m²). We were unable to distinguish whether the main trigger was emotional or orthostatic in only nine syncope subjects. They were excluded from the analysis. We identified no situational syncope following urination, defecation, swallowing, or coughing in this young population and during the interaction for questionnaire answering. There were no differences in age and BMI between the groups (Table 1).

There were more female subjects in the orthostatic syncope group compared to the control group (*p* = 0.05) but not compared to the emotional/blood phobia syncope group. There were no differences in the reaction time between groups (Table 1, congruence). The Simon effect was increased in the emotional/blood phobia syncope group in comparison to either the orthostatic syncope group or the control group (55.8 ± 15.7 ms vs. 13.4 ± 9, and 20.6 ± 4.9 ms, *p* < 0.05, Figure 1). There was a heterogeneity of variance due to the larger dispersion of the Simon effect in the emotional/blood phobia group (Figure 1).

## 4. Discussion

The major finding of our study was that emotional/blood phobia and orthostatic syncope subjects showed differences in the performance of a decisional task. Surprisingly, the Simon effect was most pronounced in the emotional/ blood phobia syncope subject group compared to the other two groups, who demonstrated similar decision-making abilities.

Studies on vasovagal syncope were limited by the nature of this phenomenon. The prevalence was high, and syncope was episodic by nature, with a clinical examination that was typically uninformative between episodes [2,3,14]. Moreover, the most critical aspect of diagnosis was the history taking, which could often establish a diagnosis on its own in typical cases [2,15,16]. The head-up tilt test could be useful in certain cases [7,15]. This test is widely used to study vasovagal syncope due to its objectivity by comparing patients with a positive and negative response to the test, as seen in examples such as [5,17,18]. However, its design is limited by the low sensitivity and specificity of the head-up tilt test [7,15]. Additionally, most studies only include patients seeking medical advice, which may bias the findings toward a particular population with vasovagal syncope (Bergson’s bias) [19]. To address these limitations, we designed our study to include outwardly healthy young individuals from the general population and based our patient classification on history taking, despite its lack of objectivity, because it remains the most crucial aspect of diagnosis [2,15,16]. Nevertheless, our study design also had limitations, which we discuss further below. The dividing line between subjects with and without syncope is not always clear-cut. We applied an original design to the study of vasovagal syncope that excluded from the analysis subjects who could not be reliably assigned to either the syncope or the control group. While phenomena other than vasovagal syncope (for example, cardiac syncope or seizure) could result in a transient loss of consciousness, their probability was assumed to be very low among young healthy subjects reporting an iterative transient loss of consciousness [1,6].

Decision-making is part of daily life, and impairments are shown in a variety of circumstances [20]. Several tools assess decision-making in psychological studies. They usually test the cognitive capacity to balance between choices to maximize profit and give time to think [20]. Survival implies fast choices and coordination with physiological behavior and stress responses. We aimed to test these fast choices, and the Simon task, based on reaction times, seemed appropriate. The Simon task is also based on conflicting sensory inputs [10,12]. A main point of our hypothesis was that the context of standing in a warm environment, which requires both vasoconstriction and vasodilation, leads to conflicting sensory inputs that increase the risk of syncope. This raised the possibility that impaired decision-making could be implicated in syncope. Our study showed altered decision-making in subjects with vasovagal syncope. Decision-making is a complex process with several steps [20]. The question remains as to which steps were altered in subjects with vasovagal syncope, and further studies are needed.

The known emotional responsivity of patients experiencing vasovagal symptoms could provide evidence for the connection between the autonomic nervous system and the brain [21]. These connections are, however, well-established [22,23,24]. The current understanding is that the automatic baroreflex loop regulates blood pressure on a beat-by-beat basis and that the brain provides an occasional input to coordinate behavioral and stress responses as needed [22]. Our hypothesis suggests that blood pressure regulation not only involves the baroreflex but also decision-making when managing conflicting sensory information. This complex view aligns with a recent concept in the field of system dynamics, which explains the stability of biological variables through self-organized criticality [25,26]. In these dynamic systems, spontaneous large events known as avalanches occur [26]. Our recent findings support the concept that vasovagal syncope is similar to avalanches and highlight the role of the brain’s higher functions in regulating blood pressure on a beat-by-beat basis [17].

The primary limitation of this study was the limited sample size of subjects with emotional/blood phobia syncope, which accounted for a larger variability in their Simon effect compared to the other groups. Initially, it was anticipated that these groups would have more equal sizes based on a prior study of a similar population (medical students) [13]. However, the goal of that study was to provide an epidemiological overview of syncope, which differed from the purpose of our study. Unlike our study, the authors of the previous study did not have to exclude a significant number of subjects to control for confounding factors. Another limitation of this study was the unequal proportion of female subjects between the control and orthostatic groups, while the literature suggests a higher proportion of females in the emotional/blood phobia group [13]. However, the proportion of female subjects was not different between the emotional/blood phobia and orthostatic groups, which is the main comparison when comparing the study hypothesis. Given the focus on psychological aspects, the inherent low reproducibility of psychological studies and methods is well-known and remains an unavoidable limitation despite efforts to standardize procedures [27]. Nevertheless, this study was performed in accordance with the standardization of the Simon task [12]. Further research with larger sample sizes is required to validate our findings and determine if there are other cerebral and cardiovascular differences between emotional/blood phobia and orthostatic syncope subjects. The self-reports of the circumstances surrounding a transient loss of consciousness during the brief questionnaire lacked objectivity, in contrast to studies that incorporate the objective results of a head-up tilt test into their design. However, the study design mitigates this issue. The target population was young individuals, and the “unclear” group was excluded from the analysis. Additionally, one of the authors, not the subjects themselves, performed the trigger classification. The brief questionnaire allowed for the inclusion of the largest possible number of subjects, and Ganzeboom et al. previously demonstrated that questionnaires could efficiently establish the history of subjects with syncope for research purposes [13]

## 5. Conclusions

Using an original study design, our research offers new insights suggesting that vasovagal syncope may involve several distinct conditions and that altered decision-making plays a role in blood phobia vasovagal syncope. This latter finding could lead to the improved management of individuals suffering from recurrent blood phobia vasovagal syncope by focusing on enhancing their decision-making abilities. However, a question remains as to whether behavioral therapy and hypnosis, which have been used to improve decision-making in other conditions, might be applicable in the context of vasovagal syncope [28].

## Figures and Tables

**Figure 1 biology-12-00930-f001:**
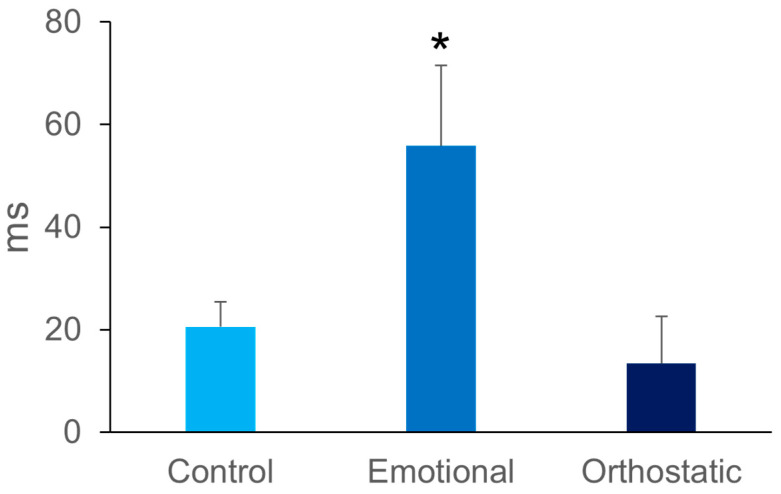
The Simon effect (y axis) in three groups of subjects: one with no syncope (control), one with an emotional/blood phobia vasovagal syncope, and one with an orthostatic vasovagal syncope. The error bars represent the standard error. * *p* < 0.05 between emotional/blood phobia and either control or orthostatic.

**Table 1 biology-12-00930-t001:** Group characteristics and performances.

	Control	Emotional	Orthostatic	
N	98	10	38	
Age (y)	22 ± 2	21 ± 2	22 ± 2	Ns
BMI (kg/m²)	22 ± 2	21 ± 2	21 ± 2	Ns
**Congruence (Non-Confusing Stimulus)**				
Reaction (ms)	440 ± 8	485 ±44	460 ± 20	Ns
Accuracy (%)	97 ± 1	95 ± 2	98 ± 1	Ns
**Incongruence (Confusing Stimulus)**				
Reaction (ms)	460 ± 8	540 ± 53	473 ± 16	*p* < 0.05
Accuracy (%)	92 ± 1	90 ± 5	94 ± 1	Ns

N: number, BMI: body mass index, Reaction: reaction time, Ns: no significant difference. Statistical differences are detailed in the text.

## Data Availability

The data presented in this study are available on reasonable request from the corresponding author.

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
