# Peer review of "Decision-Making in Patients with Vasovagal Syncope: A Preliminary Study"

_biology, 2023, doi:10.3390/biology12070930_

Round 1
Reviewer 1 Report (Previous Reviewer 1)
The authors demonstrated the difference between emotional syncope and orthostatic syncope in subjects with vasovagal syncope (VVS) using a decision conflict task. Usually, emotional syncope and orthostatic syncope are classified in the same category as vasovagal syncope. The authors revealed that emotional syncope subjects showed a delayed reaction time when faced with decisional conflict in comparison with control and orthostatic syncope subjects.
I agree with the author’s opinion that emotional syncope and orthostatic syncope are different category in VVS. Emotional syncope may more influence with mental factors. As the authors described, improvement of their decision-making process could be a key factor for management of subjects with emotional syncope. Their approach is unique and the result of this study will shed a new light on the pathophysiology in VVS.
However, the number of the study population is very limited, especially including only 10 subjects with emotional syncope. The authors mentioned as the preliminary study, further research with larger sample sizes is mandatory to confirm their hypothesis.
Minor point
Page 4, line 148, in table 1, the number of control group does not match with the manuscript: which is the correct number, 97 or 98? Please ascertain the correct number of the control subjects.
Author Response
We appreciate the time and effort that the reviewer has put into evaluating our work.
>Page 4, line 148, in table 1, the number of control group does not match with the manuscript: which is the correct number, 97 or 98? Please ascertain the correct number of the control subjects.
Thank you for noticing the typo in Table 1, which has now been corrected (line 151).
Reviewer 2 Report (New Reviewer)
Dear Authors,
First of all, I would like to congratulate you for the work done, however I have several doubts and therefore I ask you to please clarify these doubts for me:
1) I noticed that in the attached questionnaire there are several questions whose answers were not described in the results;
2) I also noticed that there are no results related to male participants, why? Were there no male participants or were all male participants excluded?
3) Why didn't you make a diagram demonstrating the exclusion criteria instead of describing them, as this would certainly facilitate both reading and understanding the information presented?
4) Could you please explain to me the criteria that were used to determine the exclusion of participants taking into account the criteria diseases and/or medications, please?
I eagerly await your return.
Kind regards,
The article is very well written, as the authors demonstrate a high capacity for technical and scientific writing and command of the English language.
Author Response
We appreciate the time and effort that the reviewer has put into evaluating our work.
>1) I noticed that in the attached questionnaire there are several questions whose answers were not described in the results;
Not all five questions about circumstances are individually reported, as the purpose of these questions was solely to classify subjects into the emotional, orthostatic, or uncertain group. The response of "yes" to the first question, "At the sight of blood, during care," or to the second question, "In cases of strong emotions or sharp pain," held equal importance in our objective. Similarly, the "yes" response to the last three questions regarding circumstances carried the same weight: "When you are tired, stressed," "In warm, confined, or overcrowded environments," and "In none of these circumstances." This point is elaborated on in Appendix A, lines 294 to 302.
>2) I also noticed that there are no results related to male participants, why? Were there no male participants or were all male participants excluded?
Our method of reporting the results may have been unclear, as the study population consisted of both males and females. To address this, we have revised the results section to provide clarity (lines 141 to 146). For instance, we now state, "57 subjects were identified as syncope cases (36 of whom were female; aged 22 ± 2 y; BMI: 21 ± 3 kg/m²)," instead of "57 subjects were identified as syncope cases (aged 22 ± 2 y; BMI: 21 ± 3 kg/m²; 36 female)."
We have revised Table 1 line 151 to eliminate this confusion. We have removed the female count from the table since it is already mentioned in the text.
>3) Why didn't you make a diagram demonstrating the exclusion criteria instead of describing them, as this would certainly facilitate both reading and understanding the information presented?
The diagram is now included as the Appendix B line 303.
4) Could you please explain to me the criteria that were used to determine the exclusion of participants taking into account the criteria diseases and/or medications, please?
To provide clarification on this matter, we included the following sentences in the Materials and Methods section (Lines 78-82): "The specific types of medication or health issues were not specified. Subjects with any health issues were excluded to minimize the risk of including syncope cases unrelated to vasovagal episodes. Additionally, subjects who were taking medication were excluded to minimize the risk of including any side effects that could manifest as dizziness or loss of consciousness."
This manuscript is a resubmission of an earlier submission. The following is a list of the peer review reports and author responses from that submission.
Round 1
Reviewer 1 Report
The authors demonstrated the difference between emotional syncope and orthostatic syncope in subjects with vasovagal syncope (VVS) using a decision conflict task. Usually, emotional syncope and orthostatic syncope are classified in the same category as vasovagal syncope. The authors revealed that emotional syncope subjects showed a delayed reaction time when faced with decisional conflict in comparison with control and orthostatic syncope subjects. I agree with the author’s opinion that emotional syncope and orthostatic syncope are different category in VVS. Emotional syncope may more influence with mental factors. As the authors described, improvement of their decision-making process could be a key factor for management of subjects with emotional syncope. Their approach is unique, and the result of this study will shed a new light on the pathophysiology in VVS. Therefore, the manuscript seems to be acceptable to the journal.
However, there are some limitations:
First. the number of the study population is limited, especially including only 10 subjects with emotional syncope.
Second, there is some concerns about the reproducibility of the Simon task.
Minor point:
Page 1, line 10: the number “57” may be a misprint of “38”?
Author Response
Common response to all three reviewers:
One common concern raised by all three reviewers was the study limitations and the need for further confirmation of the results. We fully acknowledge and appreciate this concern. Our study was based on a speculative hypothesis and employed an original design, which may explain the logical concern about the limitations of the study. It is important to note that the manuscript is submitted as a communication rather than a full article. To address the concerns raised, we have made several changes. Firstly, we have modified the title to include the phrase "a preliminary study" to clarify the nature of our work. Additionally, we have significantly expanded the discussion section to better outline the limitations of our study and provide more context for the results. Specifically, we have added nine lines to the paragraph on limitations (lines 221-229), and we have also included two new references (references 10 and 25). We believe that these changes have addressed the reviewers' concerns.
Specific responses to reviewer #1:
The study population is limited> Reviewer #3 requested information about the computation of the sample size. We have significantly improved the statistics section by adding 11 lines (lines 114-123). In this new statistics section, we mention the previous experiment by Bialystok et al [9], which used the Simon task and groups of 10 subjects (“However, Bialystok et al. obtained significant results in groups of 10 subjects using the Simon task [9]”). We believe that this section addresses the point raised by the reviewer.
Concerns about the reproducibility of the Simon task> The reviewer is correct in noting that reproducibility is an issue for any psychological study. We have now acknowledged this point in the paragraph of the discussion that focuses on limitations by adding lines 225-229 and reference [25]. We have also added a reference that addresses the reliability of the Simon task [10] (“Given the focus on psychological aspects, the inherent low reproducibility of psycho-logical studies and methods is well-known and remains an unavoidable limitation de-spite efforts to standardize procedures [25]. Nevertheless, this study was performed in accordance with the standardization of the Simon task [10]”)
Line 10: The number 57 may be a misprint of 38: The reviewer is correct. The number has been corrected.
Reviewer 2 Report
The paper focus the attention on the influence of blood phobia or orthostatic vasovagal syncope in decision-making ability. The authors analyzed 145 subjects who performed a decisional conflict task to quantify their conflict-management ability. The task was the computer version of the Simon Task. Emotional/blood phobia syncope subjects showed a delayed reaction time when faced with decisional conflict in comparison with control and orthostatic syncope subjects. The authors concluded that emotional/blood phobia and orthostatic syncope are two clinical entities.
The article is quite clear even if the results obtained in my opinion are very weak.As the authors say, a strong limitation of the study is the limited sample size of subjects with emotional/blood phobia that creates a strong imbalance in the number of subjects per group. Furthermore, there were more girls in the orthostatic syncope than control group, which could influences the results. The main question is: did the authors calculated the simple size for this study? It is not reported in the manuscript. It is necessary in order to avoid these differences between groups. The study should be remodulated consideringthe larger variability in the questionnaire answers.
I have also some other comments:
· In the abstract and in the result section the number of subjects included and excluded for the analysis are not so clear. Please write again this part
· In the materials and methods section, the authors should better explain inclusion and exclusion criteria. For example, age interval of the subject is not clear and is reported only in appendix A.
· In figure 1 the graph should be colored to better distinguish the groups. The visual effect makes the idea better to author to understand.
Author Response
Common response to all three reviewers:
One common concern raised by all three reviewers was the study limitations and the need for further confirmation of the results. We fully acknowledge and appreciate this concern. Our study was based on a speculative hypothesis and employed an original design, which may explain the logical concern about the limitations of the study. It is important to note that the manuscript is submitted as a communication rather than a full article. To address the concerns raised, we have made several changes. Firstly, we have modified the title to include the phrase "a preliminary study" to clarify the nature of our work. Additionally, we have significantly expanded the discussion section to better outline the limitations of our study and provide more context for the results. Specifically, we have added nine lines to the paragraph on limitations (lines 221-229), and we have also included two new references (references 10 and 25). We believe that these changes have addressed the reviewers' concerns.
Specific responses to reviewer #2:
This means that they experienced syncope in the past that was not investigated> It is not uncommon for a population of students aged between 18 and 30 to have a history of loss of consciousness that has not been thoroughly investigated. A single or few episodes of loss of consciousness may prompt medical advice, but it is not common for these young individuals to require regular visits to a doctor (as indicated by point P1.2 of the questionnaire). If a subject answers "Yes" to P1.2, indicating a history of loss of consciousness, but the episodes did not require follow-up or regular visits to a doctor, they were included in the study. To clarify this point, we have added the following sentence to the Material and Methods section, under the questionnaire section: "Subjects with a medical history that included transient loss of consciousness but did not require follow-up or regular visits to a doctor were included” (line 78-80).
Then how can you be certain from an objective point of view of the origin of the syncope?>. It is not always possible to be certain of the origin of syncope, as noted in the literature [6, 13]. However, epidemiological data suggest that the likelihood of other causes of loss of consciousness in young people is low. To increase the probability of a correct diagnosis in our population, we only included young people who had experienced at least three spells and had no medical treatment or issues requiring regular visits to a doctor. Nevertheless, as we added in the discussion section, there are cases where the diagnosis of vasovagal syncope cannot be confirmed even after medical evaluation [6, 13] (“However, in some cases, the diagnosis of vasovagal syncope cannot be confirmed even after medical evaluation [6, 13]”. line 169-170).
No medical data beside age and BMI was provided>. The reviewer is correct in noting that no medical or anthropological data, aside from age and BMI, was provided. However, we purposely kept the questionnaire as simple as possible to include as many subjects as possible. Conducting measurements on site would raise several concerns, such as not meeting the consensus conditions for blood pressure measurements according to the latest update we are aware of (Am J Med, Jan 6, 2023). Furthermore, the standardization of the simon task does not include blood pressure or heart rate measurement [9, 10]. For a study on loss of consciousness, blood pressure and heart rate measurements would require supine and standing position assessments for at least the first three minutes, and potentially even a whole head-up tilt test. Finally, the questionnaire was administered in-field. We have previous experiences with in-field experiments that involved measurements (Fortrat et al., Front Physiol, 2017,8:694; Ravé & Fortrat, Eur J Appl Physiol, 2016, 116:1575-82; Sigaudo-Roussel et al., Eur J Appl Physiol, 2002,86:258-65). These studies required specific ethical and administrative procedures to show that in-field measurements were the only feasible way to conduct the study. However, this was not necessary for the present study. To address this point, we added the following statement to the manuscript: "Measurements such as blood pressure could not be performed due to the inability to reach a consensus on the measurement conditions during this in-field questionnaire. Additionally, administrative and human experiment issues also contributed to this limitation" (line 88-91).
You also mention in your manuscript that the tilt test is an archaic exploration and a simple questionnaire would be superior>. The term "archaic" does not appear in our manuscript, and we understand the value of head-up tilt tests in managing orthostatic intolerance; in fact, we frequently use them with our patients. We suspect the reviewer may have misunderstood our intent due to some lack of clarity in our writing. Our goal was not to present a medical perspective or provide objective medical data; rather, we submitted our manuscript to Biology, a scientific journal, and designed the questionnaire solely for the experimental purpose of this study and to enhance our understanding of vasovagal syncope. At present, our study has no medical implications, but it does advance knowledge of vasovagal mechanisms and suggests some potential medical applications. To clarify these points, we made several changes throughout the manuscript. For example, we explicitly state on lines 65-66 that " The questionnaire was designed to achieve the study's objectives and not to propose a new medical tool." We also indicate that the questionnaire provides "a summary of" the subject's history (line 80), and we emphasize the diagnostic usefulness of the head-up tilt table test in assessing patients with transient loss of consciousness or orthostatic intolerance, as described in line 167 to 169 and supported by reference [13]. We no longer claim that patient history is the best diagnostic tool, but instead say on lines 177-178 that "The first step to diagnose vasovagal syncope is the history of the patient [13, 17]", while also noting that the Ganzeboom questionnaire was developed for research purposes (line 179).
Also when and in what frame was the study conducted?> To address this point, we included 9 additional lines in the Questionnaire section of the Material and Methods (lines 66-73). We specified that the experiment was conducted over five-week campaigns spanning five years, with the first two campaigns aimed at improving our procedures. Furthermore, we indicated that the three inclusion campaigns took place between 2015 and 2017.
Reviewer 3 Report
Dear authors, your manuscript raises an interesting idea, however I consider the way it is approached is quite superficial. Firstly, you targeted individuals that had had syncope in the past compared to individuals that had not, then you classified the syncope as orthostatic and emotional, simply based on what the test subject declared? From my understanding only healthy individuals were included, did this mean individual with no known medical history? This means they had syncope in the past which was not investigated, did I understand correctly? then how can you be certain from an objective point of view of the origin of the syncope? Also, no medical data besides age and BMI index was provided, at least blood pressure could and pulse before during and after the test could have been measured.
You also mention in your manuscript that the tilt test is a archaic exploration and a simple questionnaire would be superior, I do not find this to be very accurate from a medical stand point.
From my point of view this manuscript takes the idea of syncope quite lightly and does not offer very much objective medical data other than the presented questionnaire which does not actually cover a significant amount of medical data. Perhaps the authors would care to elaborate on the relevance of syncope and medical test that are used to identify and/or exclude actual pathological causes that may lead to syncope. Also, perhaps they could include other medical data collected from these individuals.
Also when and in what time frame was the study conducted ?
Author Response
Common response to all three reviewers:
One common concern raised by all three reviewers was the study limitations and the need for further confirmation of the results. We fully acknowledge and appreciate this concern. Our study was based on a speculative hypothesis and employed an original design, which may explain the logical concern about the limitations of the study. It is important to note that the manuscript is submitted as a communication rather than a full article. To address the concerns raised, we have made several changes. Firstly, we have modified the title to include the phrase "a preliminary study" to clarify the nature of our work. Additionally, we have significantly expanded the discussion section to better outline the limitations of our study and provide more context for the results. Specifically, we have added nine lines to the paragraph on limitations (lines 221-229), and we have also included two new references (references 10 and 25). We believe that these changes have addressed the reviewers' concerns.
Specific response to reviewer #3
The main question is: did the authors calculated the sample size?> To address this concern, we added a new paragraph consisting of 11 lines at the beginning of the statistical section (lines 114-124). As noted, we were unable to statistically calculate the sample size due to the absence of comparable data in the literature. Nonetheless, we estimated the sample size based on a previous study that utilized the Simon task [9] and the epidemiological data reported by Ganzeboom et al. [11].
There were more girls in the orthostatic syncope than control group>. We believe that the new paragraph added to the statistical section enhances the discussion on this limitation (see previous point and lines 114-124). Nevertheless, we also included a discussion of this limitation in the main section of the discussion section and mentioned “Another limitation of this study is the unequal proportion of female subjects between the control and orthostatic groups, while the literature suggests a higher proportion of females in the emotional/blood phobia group [11]. However, the proportion of female subjects was not different between the emotional/blood phobia and orthostatic groups, which is the main comparison to test the study hypothesis” (line 221-225).
In the abstract and in the results section the number of subjects included and excluded for the analysis is not so clear.> To address this issue, we revised both the abstract and the results section (lines 17-18 and 20-22 of the abstract and lines 130-133 of the results section) to improve clarity and precision.
In the materials and methods section, the authors should better explain inclusion and exclusion criteria.> The age interval of subjects is now clearly mentioned in this section and we add the following paragraph to the materials and methods section to clarify this point: “The questionnaire, which can be found in the appendix, served to identify individuals older than 30-year-old and those with health issues, and to exclude them from the analysis. A subject was considered to have a health issue if they reported needing medication or if the issue required regular visits to a doctor. Subjects who had taken medication on the day of the experiment were also excluded. Subjects with a medical history that included transient loss of consciousness but did not require follow-up or regular visits to a doctor were included” (line 73-80).
The graph should be colored.> Done.
Round 2
Reviewer 2 Report
The authors answered to reviewers' comments by adding several changes in the text that improved the manuscript.
I have two comments.
The conclusions should be improved. The authors should explain which insights into the pathophysiology they found and how they think to manage individuals with vasovagal syncope and improve their decision-making.
Regarding Figure 1. It has been colored but it should be better if you color different group using different colors.
Author Response
The conclusions should be improved.> We rewrote the conclusion. We now clearly mention the insights. We provide some directions to improve decision making. Here is the new conclusion:
Using an original study design, our research offers new insights suggesting that vasovagal syncope may involve several distinct conditions, and that altered decision-making plays a role in blood phobia vasovagal syncope. This latter finding could lead to improved management of individuals suffering from recurrent blood phobia vasovagal syncope by focusing on enhancing their decision-making abilities. However, a question remains as to whether behavioral therapy and hypnosis, which have been used to improve decision-making in other conditions, might be applicable in the context of vasovagal syncope [28].
Regarding Figure 1. > We colored Figure 1 with different colors.
Reviewer 3 Report
The authors have made several edits to the manuscript, my remaining issues would be as follows.
Firstly, if this study was done during a five-year time period up until 2017 why are the results only being published now?
“Our goal was not to present a medical perspective or provide objective medical data”
Then syncope is not a medical issue? Also how does writing scientific manuscript not require objective medical data?
Another point you made in the revision is that “The questionnaire was designed to achieve the study's objectives and not to propose a new medical tool”. I did not mention such a questionnaire could be used as a “new medical tool”, I only emphasized the fact that in my view it lacks objectivity.
However, in the first version you wrote “The best tool to diagnose vasovagal syncope is the 178 history of the patient” and then corrected it to being a first step. Is it or is it not a tool to evaluate syncope in your opinion? Because this is a rather significant modification. Then if it is not a tool to assess whether or not the patient had syncope, then this manuscript is based only on a “first step”.
Returning to the issue of blood pressure. Your reply and edits to the text made it clear that no blood pressure was measured at any time and in any way in your subjects, making it not a subject of this manuscript. However, your discussion included the following phrase “Our hypothesis suggests that blood pressure reg- 207 ulation not only involves the baroreflex, but also decision-making to manage conflicting 208 sensory information. This complex view aligns with a recent concept in the field of sys- 209 tem dynamics, which explains stability of biological variables through self-organized 210 criticality”. If blood pressure was not a factor in this particular study why discuss about it’s potential role in decision making?
Regarding your conclusion:
“Our study provides new insights into the pathophysiology of vasovagal syncope by 239 using an original study design.”
What would those new insights (plural) be, specifically?
“Our results suggest decision-making plays a role in vas- 240 ovagal syncope. This finding should now lead to a better management of individuals 241 suffering from iterative vasovagal syncope by focusing on improving their deci- 242 sion-making.”
In what ways could those individuals be better managed and how could their decision making be improved?
Author Response
Why are the results only being published now?> The delay in submitting was due to personal reasons of one of the authors, which we cannot disclose here in order to respect their privacy. Nonetheless, we have taken steps to ensure that the delay did not compromise the quality and integrity of the research findings. We apologize for any inconvenience this may have caused and appreciate your understanding.
Then syncope is not a medical issue?> Upon carefully reviewing the reviewer's second-round comments and revisiting the first-round feedback, we have come to the realization that we did not fully comprehend the reviewer's point initially. As a result, we have completely rewritten the paragraph concerning the head-up tilt test, with a focus on addressing the reviewer's concerns about the test's objectivity compared to the potential subjectivity of a questionnaire. We sincerely hope that our revisions demonstrate a deeper understanding of the reviewer's feedback and that they are satisfied with our revised submission:
Studies on vasovagal syncope are limited by the nature of this phenomenon. The prevalence is high, and syncope is episodic by nature, with a clinical examination that is typically uninformative between episodes [2, 3, 14]. Moreover, the most critical aspect of diagnosis is the history taking, which can often establish a diagnosis on its own in typical cases [2, 15, 19]. The head-up tilt test can be useful in certain cases [7, 15]. This test is widely used to study vasovagal syncope due to its objectivity, by comparing patients with a positive and negative response to the test, as seen in examples such as [5, 16, 17]. However, this design is limited by the low sensitivity and specificity of the head-up tilt test [7, 15]. Additionally, most studies only include patients seeking medical advice, which may bias the findings towards a particular population with vasovagal syncope (Bergson’s bias) [18]. To address these limitations, we designed our study to include outwardly healthy young individuals from the general population and based our patient classification on history taking, despite its lack of objectivity, because it remains the most crucial aspect of diagnosis [2, 15, 19]. Nevertheless, our study design also has limitations, which we discuss further below.
We also added the following sentence in the limitation section of the discussion:
Self-reports of the circumstances surrounding transient loss of consciousness during brief questionnaire lack objectivity, in contrast to studies that incorporate the objective results of a head-up tilt test into their design.
Returning to the issue of blood pressure.> We agree that our previous mention of this issue was incomplete. In order to address this, we have added the following sentences to this section to provide a more comprehensive explanation:
Although a tendency toward low blood pressure has been reported, the clinical examination of patients with vasovagal syncope is typically uninformative [2, 7, 11]. Furthermore, blood pressure is expected to be within the normal range for this population of young people with no health issues who receive medical care from the university.
Regarding your conclusion:> We have revised our conclusion to include a clear and detailed mention of our insights, as well as provided a reference as a suggestion to improve decision-making. Here is the new conclusion:
Using an original study design, our research offers new insights suggesting that vasovagal syncope may involve several distinct conditions, and that altered decision-making plays a role in blood phobia vasovagal syncope. This latter finding could lead to improved management of individuals suffering from recurrent blood phobia vasovagal syncope by focusing on enhancing their decision-making abilities. However, a question remains as to whether behavioral therapy and hypnosis, which have been used to improve decision-making in other conditions, might be applicable in the context of vasovagal syncope [28].
Round 3
Reviewer 2 Report
The manuscript have been improved.
I have no other comments.
Reviewer 3 Report
I would like to thank the authors for their reply, however my aforementioned concerns regarding the manuscript stand as there is very little objective and/or scientific data to go by. While I see some of it's potential at this point I find it to be incomplete.
I understand the ideas behind the study and the authors' intentions behind it, however in my view and at this stage it is still quite superficial. I noticed that you added in your conclusion that "However, a question remains as to whether behavioral therapy and hypno- 268 sis, which have been used to improve decision-making in other conditions, might be applicable in 269 the context of vasovagal syncope". Again, I would not recommend mentioning something in the conclusion that has not been remotely mentioned anywhere else in the manuscript. If you wished to include this you should have discussed it before.